# Maize Herbivore-Induced Volatiles Enhance Xenobiotic Detoxification in Larvae of *Spodoptera frugiperda* and *S. litura*

**DOI:** 10.3390/plants14010057

**Published:** 2024-12-27

**Authors:** Peng Wang, Qiyue Zeng, Yi Zhao, Xiaomin Sun, Yongqiang Han, Rensen Zeng, Yuanyuan Song, Dongmei Chen, Yibin Lin

**Affiliations:** 1Ministry of Education Key Laboratory for Genetics, Breeding and Multiple Utilization of Crop, Laboratory of Ministry of Agriculture and Rural Affairs of Biological Breeding for Fujian and Taiwan Crops, Fujian Agriculture and Forestry University, Fuzhou 350002, China; fafuwp@163.com (P.W.); zhaoyipgpr@163.com (Y.Z.); sunxiaomin2023@163.com (X.S.); rszeng@fafu.edu.cn (R.Z.); 2Fujian Provincial Key Laboratory of Crop Biotechnology, College of Agriculture, Fujian Agriculture and Forestry University, Fuzhou 350002, China; 3College of Science, Southern University of Science and Technology, Shenzhen 518055, China; 12112448@mail.sustech.edu.cn; 4College of Life Sciences and Resource Environment, Yichun University, Yichun 336000, China; hanyongqiang1984@163.com

**Keywords:** herbivore-induced plant volatiles, *Spodoptera frugiperda*, xenobiotic, cytochrome P450, DIMBOA

## Abstract

The release of herbivore-induced plant volatiles (HIPVs) has been recognized to be an important strategy for plant adaptation to herbivore attack. However, whether these induced volatiles are beneficial to insect herbivores, particularly insect larvae, is largely unknown. We used the two important highly polyphagous lepidopteran pests *Spodoptera frugiperda* and *S. litura* to evaluate the benefit on xenobiotic detoxification of larval exposure to HIPVs released by the host plant maize (*Zea mays*). Larval exposure of the invasive alien species *S. frugiperda* to maize HIPVs significantly enhanced their tolerance to all three of the well-known defensive compounds 2,4-dihydroxy-7-methoxy-1,4-benzoxazin-3-one (DIMBOA), chlorogenic acid, and tannic acid in maize and the two commonly used insecticides methomyl and chlorpyrifos. HIPV exposure also improved the larval tolerance of *S. litura* third instars to chlorogenic and tannic acids. Furthermore, larval exposure to either maize HIPVs or DIMBOA induced the activities of cytochrome P450 enzymes (P450s), glutathione-s-transferase (GST), and carboxylesterase (CarE) in the midguts and fat bodies of the two insects, while the induction was significantly higher by the two components together. In addition, the expression of four genes encoding uridine diphosphate (UDP)-glycosyltransferases (*UGT33F28*, *UGT40L8*) and P450s (*CYP4d8*, *CYP4V2*) showed similar induction patterns in *S. frugiperda.* Cis-3-hexen-1-ol, an important component in maize HIPVs, also showed the same functions as maize HIPVs, and its exposure increased larval xenobiotic tolerance and induced the detoxification enzymes and gene expression. Our findings demonstrate that HIPVs released by the pest-infested host plants are conductive to the xenobiotic tolerance of lepidopteran insect larvae. Hijacking the host plant HIPVs is an important strategy of the invasive alien polyphagous lepidopteran pest to counter-defend against the host plant’s chemical defense.

## 1. Introduction

Plants and insects contribute the majority of the biodiversity on Earth. During the long history of the coevolution of the two major groups of organisms, plants have taken advantage of the easy synthesis of organic compounds to produce numerous toxic secondary metabolites to defend against insect herbivores. Upon herbivore attack, plants perceive damage-associated and herbivore-associated molecular patterns and immediately activate early signaling components such as Ca^2+^, reactive oxygen species (ROS), and MAP kinases. Subsequently, plants initiate their signaling networks including the activation of phytohormones and transcription factors, leading to transcriptional reprogramming and a series of metabolic, physiological, and biochemical changes including the increased production of secondary metabolites [1].

During co-evolution with insect herbivores, plants have developed both constitutive and inducible defenses at multiple morphological, molecular, and biochemical levels [2,3,4]. The production of defensive secondary metabolites such as 2,4-dihydroxy-7-methoxy-1,4-benzoxazin-3-one (DIMBOA), diterpenoid glycosides, and pyrethrins is a key strategy for plant defense against insect herbivores [5,6,7]. In response, herbivorous insects have evolved intricate strategies to evade the toxicity of defensive compounds produced by host plants, including chelation, excretion, metabolic degradation, and target resistance mutations [8]. The main detoxification enzymes in insects are cytochrome P450 monooxygenase (P450), UDP-glucuronide transferase (UGT), glutathione-S-transferase (GST), and carboxylesterase (CarE), which play vital roles in the development of insect metabolic resistance to xenobiotics including various phytochemicals and synthetic insecticides [8,9].

Furthermore, upon insect herbivory, plants rapidly synthesize and release a complex blend of volatile chemicals named herbivore-induced plant volatiles (HIPVs) to either directly repel and intoxicate the enemies or indirectly attract the natural enemies of insect herbivores [10,11,12]. HIPVs, mainly consisting of green leaf volatiles, terpenes, and aromatic compounds, are important transmitters of plant communication with other organisms in the environment. Timely emission of HIPVs acts as a key strategy of plant adaptation to insect herbivory. Importantly, the HIPVs can serve as important agents for the induction and priming of plant defense against insect pests, showing potential in the management of agricultural pests [13,14,15]. However, little is known about the counter defense of insect herbivores in response to plant HIPVs.

The fall armyworm *Spodoptera frugiperda* (Lepidoptera) is a new alien invasive insect pest in Asia from the Americas [16]. The insect is a highly polyphagous lepidopteran pest with more than 300 host plants. It has rapidly spread in new regions and become one of the most destructive pests due to its broad host range, high reproductive potential, and swift migration [17]. The recently published genomic data of *S. frugiperda* showed that the P450 gene family is notably expanded, with 425 members, of which 283 are unique in *S. frugiperda* when compared to its related species, *Spodoptera litura,* a native polyphagous lepidopteran pest in Asia [17]. This expansion may confer *S. frugiperda* with the capacity to exploit plant volatiles to augment its detoxification.

Notably, *UGT33F28* and *UGT40L8* have been identified as pivotal genes encoding glycosyltransferases involved in the detoxification of DIMBOA, the major defensive chemical in maize and other cereals [18]. The upregulation of *UGT33F28* and *UGT40L*8 has been demonstrated to bolster the detoxification capabilities of *S. frugiperda*.

Herbivorous insects that feed on plants have the ability to stimulate the production of volatile compounds in plants, known as herbivore-induced plant volatiles (HIPVs) [19,20]. HIPVs are typically complex mixtures of compounds derived from various biosynthetic pathways, primarily including terpenes, green leaf volatiles, and aromatic compounds [21]. There is evidence suggesting that HIPVs from different biosynthetic sources can elicit resistance and immunity to parasites and pathogens in lepidopteran pests [20]. Notable active volatiles identified include cis-3-hexen-1-ol, indole, β-basil, and β-farnesene [22,23,24]. Furthermore, recent research has demonstrated that HIPVs can induce adaptation to tomato chemical defenses in *S. litura* [25]. Moreover, tolerance to chemical insecticides can also be induced in *Helicoverpa armigera* [26]. However, there is no report on the novel ecological adaptation strategies of *S. frugiperda* mediated by HIPVs.

In this study, we conducted a systematic evaluation of the impact of maize HIPVs and their key component cis-3-hexen-1-ol on the larval detoxification of the main defensive compounds in maize plants and the insecticides methomyl and chlorpyrifos in *S. frugiperda* and *S. litura*. Our findings propose a novel mechanism for the invasion of *S. frugiperda*. This mechanism involves the olfactory detection of HIPVs, which induces resistance in *S. frugiperda* to DIMBOA-mediated chemical defenses in maize, thereby facilitating its invasion.

## 2. Materials and Methods

### 2.1. Plants and Insects

Seeds of maize cv. Zhengdan 958 obtained from Shandong Luyan Agricultural Seed Co., Ltd. (Jinan, China) were disinfected using a 2% NaClO solution for 10 min and then placed in peat soil (German Dahan type: 413, particle size: 0–6 mm) in a greenhouse at 28 °C with 60% relative humidity under a 16 h light (200 μmol m^−2^ s^−1^) and 8 h dark photoperiod for germination. After 7 days, uniform and healthy seedlings were chosen for transplantation for subsequent experiments. The maize plants were cultivated in a greenhouse (14 h light/10 h dark) with 70% relative humidity and a temperature regimen of 30 °C during the day and 25 °C at night. The maize plants were adequately irrigated with a low-phosphorus Hoagland nutrient solution. The initial population of *S. frugiperda* was provided by Professor Lin Jintian at Zhongkai University of Agriculture and Engineering. The insects were reared for more than 10 generations on artificial diets as described by Gupta et al. [27]. The moths were nurtured in a 10% honey water solution and reared at constant environmental conditions (25 ± 2 °C, 60% relative humidity, light–dark = 16:8 h).

### 2.2. Larval Exposure to Maize HIPVs and Its Effects on Xenobiotic Tolerance

A directed airflow apparatus, shown in Appendix A, was used to examine the effects of exposure to maize HIPVs on the performance of 3rd and 4th instar larvae of *S. litura* and *S. frugiperda* on toxin-contained diets with 1 μg·g^−1^ DIMBOA, 3 μg·g^−1^ chlorogenic acid (CA), and 2 μg·g^−1^ tannic acid (TA). Ten maize seedlings at the five-leaf stage (15 days old) were transplanted in a sealed box (60 cm in length, 48 cm in width, and 55 cm in height). Fourth instar larvae of *S. frugiperda* and *S. litura* were inoculated on the maize seedlings and allowed to feed on the plants for 24 h. Subsequently, four treatment groups were established, including *S. litura* − HIPVs, *S. litura* + HIPVs, *S. frugiperda* − HIPVs, and *S. frugiperda* + HIPVs. This experiment was triplicated. The 3rd and 4th instar larvae of the two species were exposed to maize HIPVs and reared on diets either without toxins or with toxic defensive compounds. Weight gain of the 3rd and 4th instar larvae following exposure to maize HIPVs and feeding on an artificial diet for 24 h was measured. The susceptibility of the 3rd and 4th instar larvae to phytotoxins including chlorogenic acid, tannic acid, and DIMBOA, as well as the two common insecticides chlorpyrifos and methomyl was assessed. Finally, the tolerance of 4th instar *S. frugiperda* larvae to the two insecticides chlorpyrifos (8000 μg·mL^−1^) and methomyl (500 μg·mL^−1^), two commonly utilized broad-spectrum insecticides [28,29], was evaluated after 24 h of exposure to maize HIPVs. To determine the role of P450 in the HIPV-enhanced larval tolerance to insecticides, the 4th instar larvae of *S. frugiperda* were exposed to HIPVs for 24 h and topically treated with piperonyl butoxide (PBO, a general inhibitor of P450 enzymes) on the thorax and abdomen 1 h prior to transfer to a diet containing either 500 μg·mL^−1^ methomyl or 8000 μg·mL^−1^ chlorpyrifos. The mortality after 24 h of insecticide exposure was counted.

Furthermore, the effects of HIPV exposure on the egg hatching rate, pupation rate, and emergence rate of *S. litura* and *S. frugiperda* were examined. Additionally, the activities of P450, GST, and CarE enzymes, as well as the expression levels of *UGT33F28*, *UGT40L8*, *CYP4d8*, *CYP6B6*, and *CYP4V2* in the midgut and fat body of 4th instar larvae of *S. litura* and *S. frugiperda* were quantified in the presence and absence of maize HIPVs, as well as with or without 1 μg·g^−1^ DIMBOA in the diet.

### 2.3. Larval Exposure to cis-3-hexene-1-ol and Its Effects on Xenobiotic Tolerance

To determine the specific volatile compounds responsible for changing larval tolerance, cis-3-hexen-1-ol (C_6_H_12_O, CAS:928-96-1), which is present in maize HIPVs, was selected [22,23,24]. The compound was purchased from Shanghai Macklin Biochemical Co., Ltd. (Shanghai, China). The concentrations of cis-3-hexene-1-ol were used based on the report by Abhinav et al. [30]. The dietary supplementation concentrations of DIMBAO and chlorogenic acid (CA) were 1.0 μg·g^−1^ and 3.0 μg·g^−1^, respectively. Larvae of the fourth instar of *S. litura* and *S. frugiperda* were exposed to cis-3-hexene-1-ol as shown in Appendix A. The mortality of the larvae was assessed after 24 h of feeding, with 10 larvae per group and 5 groups for each concentration of different substances (n = 50).

A head-space volatile release apparatus, shown in Appendix A, was developed to evaluate the effects of volatile compounds on the insect detoxification of xenobiotics. The volatile compound cis-3-hexen-1-ol was added in glass wool within a 2 mL sample vial. Subsequently, an 18# needle was utilized to puncture the rubber spacer on the blue cap of the sample bottle and the lid of the 3.5 L transparent bowl box to apply the volatile compounds to *S. litura* and *S. frugiperda* larvae at a specific release rate through a needle connection. Artificial diets containing 1 μg·g^−1^ DIMBOA and 3 μg·g^−1^ CA were placed in the bowl, and the insect larvae were reared on these diets for 24 h. Thereafter, the midgut and fat body were dissected, and the activities of the P450, GST, and CarE enzymes and expression levels of *UGT33F28* and *UGT40L8* in the midguts and fat bodies were evaluated.

### 2.4. Determination of Pupation Rate, Emergence Rate, and Egg Hatchability

The same number of larvae at the pre-pupal stage of *S. litura* and *S. frugiperda* were placed in containers constructed from polypropylene (PP) material. The base of the container was filled with fine sand containing 10% water. The larvae were subjected to maize herbivore-induced plant volatiles (HIPVs) (+HIPVs) and control conditions (−HIPVs). Upon completion or failure of pupation, as well as emergence or mortality, the pupation rate and emergence rate were determined (n = 100).

A delicate brush was utilized to evenly disperse the eggs deposited by *S. litura* and *S. frugiperda*. Subsequently, the dispersed eggs were placed on a sponge, which was then positioned on a square dish. The eggs were exposed to maize HIPVs (+HIPVs) and control conditions (−HIPVs). Following 96 h of exposure, the number of hatched insects was tallied (n = 100).

### 2.5. Enzyme Activity of P450, GST, and CarE

The enzyme activities of P450, GST, and CarE in insect midguts and fat bodies were determined according to the methods of Sun et al. and Tang et al. [31,32]. The midgut and fat body tissues from *S. litura* and *S. frugiperda* were used for assaying the activities of the detoxification enzymes. The tissues were dissected in PBS, then ground by homogenizing, and were centrifuged at 4 °C and 10,000× *g* for 20 min. The supernatant was immediately transferred to assay the activities of detoxification enzymes. The activities of P450, GST, and CarE were measured by using a microplate analyzer with enzyme activity assay kits (Jiangsu Yutong Biological Technology Co., Ltd., Nanjing, China) according to the manufacturer’s instructions.

### 2.6. Gene Expression Analysis

The procedures used for RNA extraction and reverse transcription of plant samples were carried out as previously described [33], with slight modifications. Total RNA was extracted from ~0.1 g of flash-frozen, powdered root samples using the Eastep^®®^ Super Total RNA Extraction kit (Promega Biotech Co., Ltd., Beijing, China) according to the manufacturer’s instructions. Total RNA was treated with RNase-Free DNaseI (TIANGEN Biotech Co., Ltd., Beijing, China), and 1 mg of total RNA was pipetted for cDNA synthesis using the GoScript Reverse Transcription System (Promega Biotech Co., Ltd., China). Real-time PCR was performed using the MonAmp ChemoHS qPCR Mix (High Rox) Kit (Monad Biotech Co., Ltd., Shanghai, China). The reaction conditions for thermal cycling were 95 °C for 5 min, followed by 40 cycles of 95 °C for 10 s, 55–65 °C for 10 s, and 72 °C for 30 s. Fluorescence data were collected during the cycle at 72 °C. The gene expression level was normalized using the *S. frugiperda* housekeeping gene GAPDH and the 2^−ΔΔCT^ method. The gene-specific primers used in this research are listed in Appendix A. Biological triplicates with technical duplicates were performed.

### 2.7. Statistical Analysis

Data were processed and plotted using Microsoft Excel 2013 and GraphPad Prism 9 software, and significance was tested using SPSS 19. All experiments were conducted using a completely randomized experimental design. The Shapiro–Wilk normality test and Levene’s homogeneity of variance test (*p* > 0.05) were used to test the normality of the data (*p* > 0.05) before all statistical analyses. Under the assumption of normality and homogeneity of variance, Student’s *t*-test or analysis of variance (ANOVA) was used to compare the differences between two or all treatments (Tukey’s test, *p* < 0.05). In the case of two-factor variance, if there was an interaction between factor 1 and factor 2 (that is, *p* < 0.05), one-way ANOVA (Tukey’s test, *p* < 0.05) was used for all treatment groups. If there was no interaction between factor 1 and factor 2 (*p* > 0.05), then Student’s *t*-test was performed for different levels of factor 1 or factor 2.

## 3. Results

### 3.1. Maize HIPVs Promote Larval Tolerance to Plant Defensive Chemicals

Without toxins in the diet, maize HIPV exposure did not affect the larval growth of *S. frugiperda* but significantly reduced the larval growth of *S. litura* (Figure 1A,B)*,* suggesting that *S. frugiperda* larvae are more adaptive to maize HIPVs. When the larvae were exposed to toxin-containing diets, maize HIPV exposure significantly improved the larval growth and toxin tolerance of both third and fourth instar larvae of *S. frugiperda* to all three tested plant defensive chemicals. Specifically, HIPV exposure increased the weight gain of third and fourth instars of *S. frugiperda* in the presence of 1 μg·g^−1^ DIMBOA by 62.8% and 60.6%, respectively (Figure 1C,F). HIPV exposure increased weight gain by 23.8% and 209.5%, respectively, in the presence of 3 μg·g^−1^ CA (Figure 1D,G), and 64.77% and 67.75%, respectively, in the presence of 2 μg·g^−1^ TA (Figure 1E,H).

For *S. litura* larvae, maize HIPV exposure did not improve the larval growth of both third and fourth instars in the presence of 1 μg·g^−1^ DIMBOA (Figure 1C,F). It did not improve the larval growth of fourth instars either in the presence of CA or TA (Figure 1G,H). However, HIPV exposure did improve the larval growth of third instars in the presence of CA and TA (Figure 1D,E,G,H). The results also suggest that *S. frugiperda* is more adaptive to maize HIPVs for tolerance to plant defensive chemicals.

### 3.2. Maize HIPVs Promote Larval Tolerance to Methomyl and Chlorpyrifos Insecticides

The susceptibility of fourth instars of *S. litura* and *S. frugiperda* fed on diets containing 500 μg·mL^−1^ methomyl or 8000 μg·mL^−1^ chlorpyrifos was examined after larval exposure to maize HIPVs for 24 h (Figure 2A,B). The mortality of fourth instar *S. frugiperda* exposed to maize HIPVs was significantly lower than that of unexposed individuals. However, no significant difference was observed between HIPV-exposed and unexposed fourth instars of *S. litura*.

In the absence of the inhibitor PBO, maize HIPV exposure reduced the larval mortality of methomyl-treated *S. frugiperda* by 43.3% and chlorpyrifos-treated *S. frugiperda* by 30.5%. In the presence of PBO, the larval mortality of insecticide-treated *S. frugiperda* significantly increased (Figure 2C,D). However, in the presence of PBO, maize HIPV exposure did not change larval mortality for both methomyl-treated and chlorpyrifos-treated *S. frugiperda*. The results indicate that P450s play a key role in the HIPV-enhanced larval tolerance to insecticides.

### 3.3. Maize HIPV Exposure Does Not Affect Insect Development

As shown in Figure 3, there was no significant difference in the egg hatching rate, pupation rate, and emergence rate between HIPV-exposed and unexposed *S. litura* and *S. frugiperda* (Figure 3A–C).

### 3.4. Maize HIPVs and DIMBOA Show Synergistic Effect on Induction of Detoxification Enzymes

We examined the activities of the three detoxification enzymes including P450, GST, and CarE in the midguts and fat bodies of fourth instar larvae of *S. litura* and *S. frugiperda* following exposure to maize HIPVs and the plant defensive chemical DIMBOA (Figure 4). Either HIPV exposure or treatment with DIMBOA significantly enhanced the activities of P450 and CarE in both the midguts and fat bodies (Figure 4A–D,I–L). More importantly, simultaneous treatments with HIPVs and DIMBOA showed the strongest induction of all three tested detoxification enzymes (Figure 4A–L). Although HIPV exposure and treatment with DIMBOA showed lower or no obvious induction of GST, simultaneous treatments with HIPVs and DIMBOA also induced the activity of GST (Figure 4E–H).

### 3.5. Maize HIPVs and DIMBOA Show Synergistic Effect on Induction of Detoxification-Associated Genes

Maize HIPV exposure led to 5.0-, 23.8-, 19.2-, and 2.2-fold upregulation of the gene expression levels of *UGT33F28*, *UGT40L8*, *CYP4d8,* and *CYP4V2* in the midguts, respectively (Figure 5A–E). The transcript levels of *UGT33F28*, *UGT40L8,* and *CYP4d8* in the midguts increased by 2.5-, 32.2-, and 4.0-fold, respectively, 24 h after feeding on a DIMBOA-containing diet. More strikingly, simultaneous treatments with HIPVs and DIMBOA showed the strongest induction of all five tested detoxification genes in the midgut (Figure 5A–E), as well as fours genes in the fat body (Figure 5G–J). The induction of *UGT33F28*, *UGT40L8*, *CYP4d8*, *CYP6B6,* and *CYP4V2* by the two components was 13.1-, 43.2-, 32.8-, 2.1-, and 3.1-fold in the midgut relative to untreated control, respectively (Figure 5A–E). In the fat body, *CYP4d8* and *CYP6B6* expression levels were induced 22.1- and 13.9-fold by simultaneous treatment with HIPVs and DIMBOA, respectively, relative to untreated control (Figure 5H,I).

### 3.6. Larval Exposure to cis-3-hexen-1-ol Enhances Tolerance to Plant Defensive Chemicals

The amount of *cis*-3-hexen-1-ol (cis-HXO) is increased in *S. frugiperda*-damaged maize plants [34]. To identify the specific volatile compounds emitted from herbivore-infested maize plants that triggered xenobiotic resistance in *S. frugiperda*, the fourth instar larvae of *S. litura* and *S. frugiperda* were exposed to the volatile cis-HXO and reared on diets contained DIMBOA and CA. We found that exposure of *S. frugiperda* larvae to cis-3-hexen-1-ol significantly enhanced larval tolerance to both DIMBOA and CA (Figure 6A,B).

### 3.7. Cis-3-Hexen-1-ol and DIMBOA Show Synergistic Effect on Induction of Detoxification Enzymes

In *S. frugiperda*, either HIPV exposure or treatment with DIMBOA significantly enhanced the activities of P450 and CarE in both the midgut and fat body (Figure 7B,D,J,L). HIPV exposure and treatment with DIMBOA only did not show induction of GST in the midgut of *S. frugiperda* (Figure 7F) but induced GST in the fat body (Figure 7H).

In *S. litura,* exposure to cis-3-hexen-1-ol induced the activities of the three detoxification enzymes P450, GST, and CarE in the fat body (Figure 7C,G,K), only induced CarE in the midgut (Figure 7I) but not P450 or GST in the midgut (Figure 7A,E). The diet supplement with DIMBOA significantly enhanced the activities of P450, GST, and CarE in both the midgut and fat body (Figure 7A,C,E,G,I,K).

Similar to the results from exposure to maize HIPVs and supplementation with DIMBOA, simultaneous treatments with cis-3-hexen-1-ol and DIMBOA showed the strongest induction of P450, GST, and CarE in both the midguts and fat bodies in the two insect species.

### 3.8. Cis-3-Hexen-1-ol and DIMBOA Upregulate UGT33F28 and UGT40L8

We further investigated the impact of exposure to cis-3-hexen-1-ol and DIMBOA treatments on the expression of two genes encoding the phase II detoxification enzymes *UGT33F28* and *UGT40L8* in in the midguts and fat bodies of fourth instar *S. frugiperda* larvae. We found that after exposure to cis-3-hexen-1-ol, the gene expression levels of *UGT3F28* and *UGT40L8* were increased by 5.4- and 3.5-fold in the midgut (Figure 8A,B), and 1.9- and 1.1-fold in the fat bodies (Figure 8C,D), respectively, in comparison to unexposed controls. Meanwhile, the diet supplemented with DIMBOA increased the gene expression levels of *UGT3F28* and *UGT40L8* by 6.7- and 1.8-fold in the midguts (Figure 8A,B), and 1.5- and 1.7- fold in the fat bodies (Figure 8C,D), respectively relative to untreated controls. Simultaneous treatments with cis-3-hexen-1-ol and DIMBOA showed the strongest induction of the two detoxification genes (Figure 8A–D).

## 4. Discussion

Currently, over 1700 volatile compounds have been identified from approximate 90 plant families globally. The majority of these compounds fall into categories such as hydrocarbons, alcohols, aldehydes, ketones, esters, and terpenes, with a molecular weight ranging from 100 to 200 [35]. It has been demonstrated that HIPVs display diverse ecological functions [36]. The HIPV-mediated interactions among plants, phytophagous insects, and natural enemies has garnered significant attention in recent years [37,38].

HIPVs have been found to have various effects on phytophagous insects, including growth inhibition, feeding deterrent, and repelling [12,39]. Additionally, HIPVs can serve as an attractor for predatory or parasitic natural enemies, thereby indirectly protecting host plants [40,41]. Despite an array of studies on the benefits of HIPVs to plants, a very limited number of studies have touched the potential benefits of HIPVs on receiver insect herbivores [29]. One obvious benefit is that insects can use HIPVs to orientate host plants. For instance, females of *Tuta absoluta* utilize HIPVs emitted by tomato plants to identify the host for oviposition [42]. A recent study revealed that symbiotic bacteria present in *Acyrthosiphon pisum* can play a role in suppressing the emission of HIPVs in host plants to reduce the recruitment of natural enemies, leading to the enhanced adaptability of herbivorous insects [43]. This study reveals that larval exposure to maize HIPVs increased the weight gain of both third and fourth instars of *S. frugiperda* in the presence of 1 μg·g^−1^ DIMBOA, 1 μg·g^−1^ CA, and 1 μg·g^−1^ TA compared to those without exposure to HIPVs. The exposure only increased the weight gain of third instars of *S. litura* in the presence of CA and TA; it did not show obvious effects on fourth instars of *S. litura* and in the presence of DIMBOA (Figure 1). Furthermore, maize HIPVs also increased the larval tolerance of *S. frugiperda* to the two insecticides but did not show an effect on the tolerance of *S. litura*. These results indicate that maize HIPV exposure showed significantly more benefits to *S. frugiperda* than to *S. litura*. Hijacking the host plant HIPVs seems to be an important strategy of the invasive alien species fall armyworm to counter-defend against the host plant’s chemical defense. More importantly, maize HIPVs and the main defensive chemical DIMBOA showed a synergistic effect on the induction of detoxification enzyme systems and detoxification genes, suggesting that the two highly polyphagous lepidopteran pests utilize both volatile HIPVs and non-volatile defensive chemicals from host plants to develop counter-defense against the host plant’s chemical defense.

Phytophagous insects have developed various mechanisms to counteract plant defenses, such as detoxification enzyme systems, physiological tolerance, and behavioral escape [8,44]. The detoxification enzymes in insects, including cytochrome oxidase (P450), carboxylesterase (CarE), and glutathione S-transferase (GSTs), play a crucial role in metabolizing plant defense substances and are essential for insect adaptation to host plant defense [9,45,46,47]. This study revealed that exposure to maize HIPVs can lead to a significant increase in the activity of detoxification enzymes in the midgut and fat body of both *S. frugiperda* and *S. litura*. However, the induction of detoxification enzymes was significantly higher in *S. frugiperda* than in *S. litura*. Furthermore, exposure to HIPVs was found to induce larval tolerance to methomyl and chlorpyrifos in *S. frugiperda* larvae but not in *S. litura*. Consequently, it is plausible to infer that *S. frugiperda* possesses the capability to promptly detect changes in maize HIPVs compared to *S. litura* and to subsequently respond physiologically to HIPVs by activating its own detoxification system to overcome the host resistance mediated by DIMBOA in maize.

Green leaf volatiles (GLVs) are a group of small gaseous molecules emitted by plants in response to various stressors such as mechanical damage, pathogen infection, and insect infestation. Cis-3-hexen-1-ol, identified as a main GLV in maize, plays a pivotal role in enhancing plant stress resistance [48]. This study found that exposure to cis-3-hexene-1-ol led to increased detoxification enzyme activities and *UGT3F28* and *UGT40L8* expression in the midgut and fat body of *S. frugiperda*. Our findings suggest that cis-3-hexen-1-ol plays a significant role in triggering the xenobiotic tolerance of *S. frugiperda* to DIMBOA-mediated chemical defense in maize. This suggests that the green leaf volatile is beneficial to both plants and insect herbivores. The application of cis-3-hexen-1-ol in enhancing plant stress resistance may confer herbivore xenobiotic tolerance.

## 5. Conclusions

Upon larval exposure to maize HIPVs, two highly polyphagous lepidopteran pests, *S. frugiperda* and *S. litura,* significantly enhanced their tolerance to plant defensive chemicals including two general defensive chemicals CA and TA and one specific defensive chemical DIMBOA. *S. frugiperda* showed more adaptiveness to maize HIPVs and gained more benefits. Larval HIPV exposure also enhanced their tolerance to two insecticides in *S. frugiperda,* but not in *S. litura.* Larval exposure to maize HIPVs also enhanced their activities of P450s, GST, and CarE in the midgut and fat body of the two insects, and the induction was significantly higher in *S. frugiperda* than in *S. litura,* which may contribute to more tolerance to xenobiotics in *S. frugiperda.* The green leaf volatile cis-3-hexen-1-ol acts as an active component in maize HIPVs to enhance larval tolerance to xenobiotics and to induce insect detoxification enzymes.

## Figures and Tables

**Figure 1 plants-14-00057-f001:**
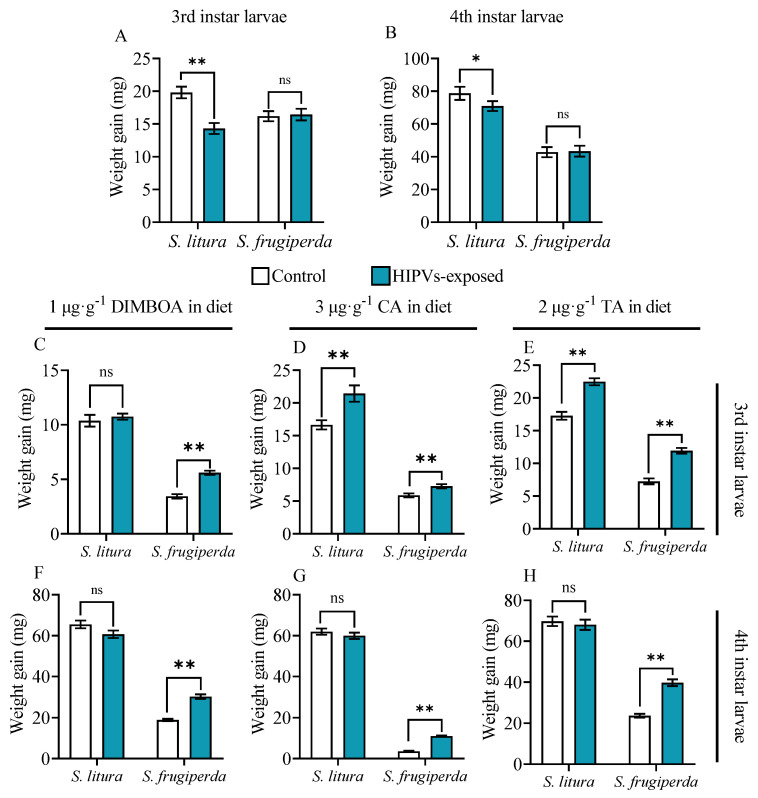
Weight gain of 3rd and 4th instar larvae of *S. frugiperda* and *S. litura* fed on artificial diets containing DIMBOA, chlorogenic acid, and tannic acid with or without exposure to maize HIPVs. The larvae were exposed to HIPVs released from living maize plants as shown in Appendix A. Weight gain of 3rd and 4th instar larvae fed on artificial diets without toxin addition (**A**,**B**), containing 1 μg·g^−1^ DIMBOA diet (**C**,**F**), 3 μg·g^−1^ chlorogenic acid (**D**,**G**), and 2 μg·g^−1^ tannic acid (**E**,**H**) for 48 h. Data are mean ± SE (n = 25). Asterisks indicate significant differences in comparison with control (Student’s *t*-test, * *p* < 0.05, ** *p* < 0.01, ns, *p* > 0.05).

**Figure 2 plants-14-00057-f002:**
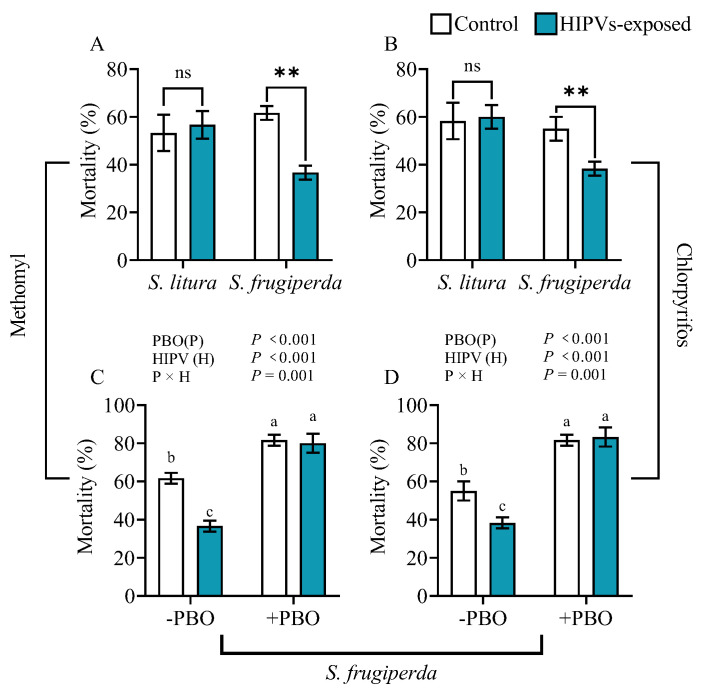
Insecticide tolerance of *S. frugiperda* and *S. litura* larvae to methomyl and chlorpyrifos after exposure to maize HIPVs. The larvae were exposed to HIPVs released from living maize plants as shown in Appendix A. Larval mortality was measured 24 h after exposure to maize HIPVs and methomyl (**A**) or chlorpyrifos (**B**). Asterisks indicate significant differences in comparison with unexposed control (Student’s *t*-test, ** *p* < 0.01, ns, *p* > 0.05). Tolerance to methomyl (**C**) and chlorpyrifos (**D**) after exposure of *S. frugiperda* larvae to maize HIPVs and the insecticide synergist piperonyl butoxide (PBO). Data are mean ± SE (n = 20). Different letters indicate significant differences (*p* < 0.05) between different treatments (Two-way ANOVA).

**Figure 3 plants-14-00057-f003:**
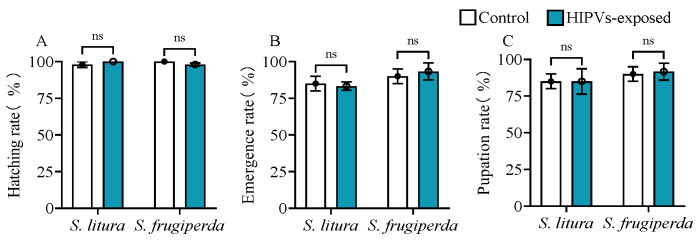
Effects of larval exposure to maize HIPVs on the developmental process of *S. litura* and *S. frugiperda*. (**A**) Egg hatching rate, (**B**) pupation rate, and (**C**) emergence rate. Data are mean ± SE (n = 100). Asterisks indicate significant differences in comparison with unexposed control (ns, *p* > 0.05).

**Figure 4 plants-14-00057-f004:**
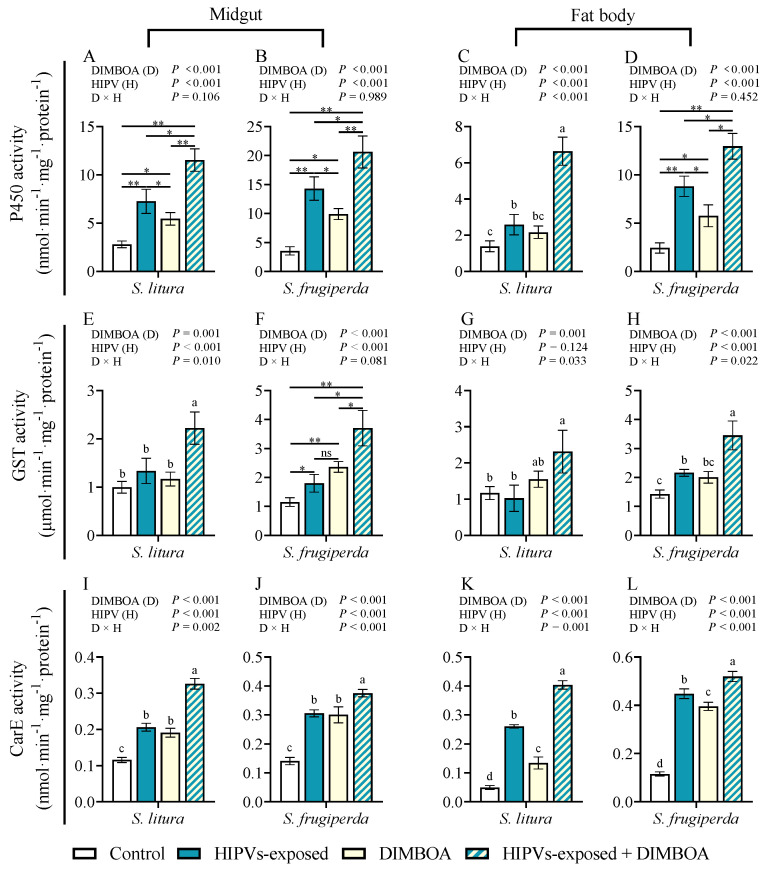
Activities of cytochrome P450 enzymes (P450s), glutathione-s-transferase (GST), and carboxylesterase (CarE) in the midguts and fat bodies of 4th instars of *S. litura* and *S. frugiperda* after exposure to maize HIPVs and feeding on DIMBOA-containing diets. P450s activities in the midguts (**A**,**B**) and fat bodies (**C**,**D**), GST activities in the midguts (**E**,**F**) and fat bodies (**G**,**H**), and CaeE activities in the midguts (**I**,**J**) and fat bodies (**K**,**L**) of *S. litura* and *S. frugiperda,* respectively. The larvae were exposed to maize HIPVs and 1 μg·g^−1^ DIMBOA diet for 24 h. Tissues dissected from five larvae were pooled, and four biological replicates were run for each treatment. Data are mean ± SE (n = 4). Asterisks indicate significant differences in comparison with control (Student’s *t*-test if the interaction between HIPVs and DIMBOA was not significant, * *p* < 0.05, ** *p* < 0.01). Different letters above bars indicate significant differences among treatments (*p* < 0.05) according to two-way ANOVA with Tukey’s multiple comparison test (if the interaction between HIPVs and DIMBOA was significant).

**Figure 5 plants-14-00057-f005:**
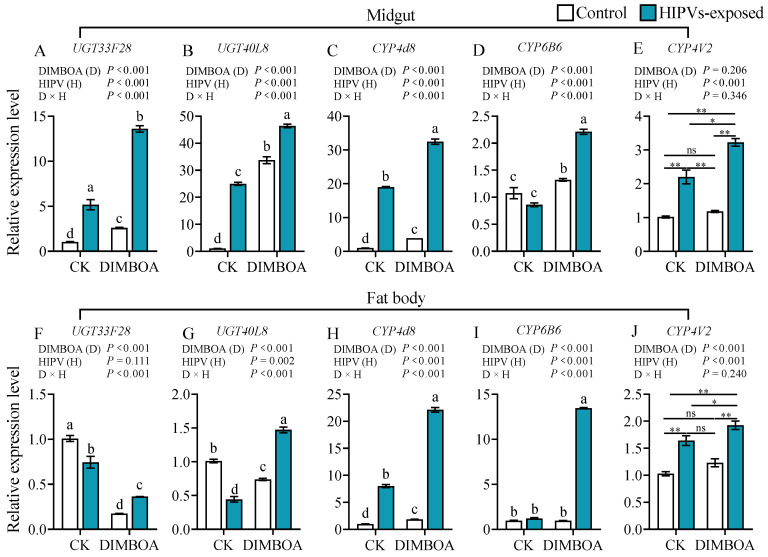
Transcript levels of *UGT33F28*, *UGT40L8*, *LOC118270728*, *CYP4d8,* and *CYP4V2* in the midguts and fat bodies of 4th instars of *S. frugiperda* after exposure to maize HIPVs and feeding on DIMBOA-containing diet. The larvae were exposed to maize HIPVs and 1 μg·g^−1^ DIMBOA diet for 24 h. Tissues dissected from five larvae were pooled, and three biological replicates were run for each treatment. Data are mean ± SE (n = 20). Asterisks indicate significant differences in comparison with control (Student’s *t*-test if the interaction between HIPVs and DIMBOA was not significant, * *p* < 0.05, ** *p* < 0.01, ns, *p* > 0.05). Different letters above bars indicate significant differences among treatments (*p* < 0.05) according to two-way ANOVA with Tukey’s multiple comparison test (if the interaction between HIPVs and DIMBOA was significant).

**Figure 6 plants-14-00057-f006:**
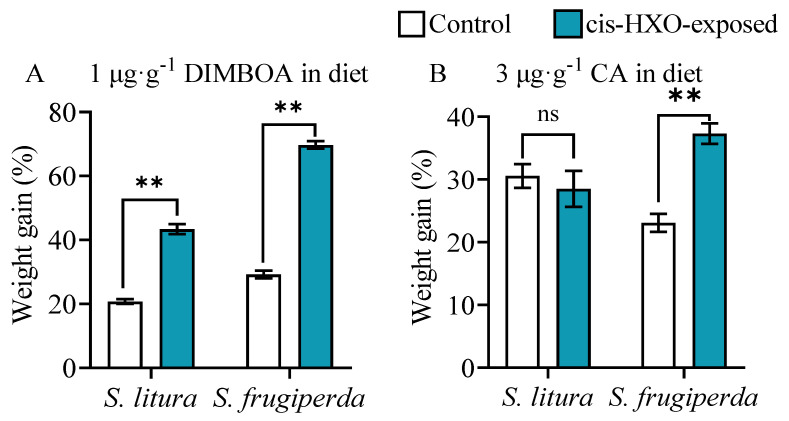
Weight gain of 4th instars of *S. litura* and *S. frugiperda* fed on diets containing 1 μg·g^−1^ DIMBOA (**A**) and 3 μg·g^−1^ chlorogenic acid (CA) (**B**) after exposure to volatile compound cis-3-hexen-1-ol (cis-3-HXO). The larvae were exposed to volatile cis-3-HXO as shown in Appendix A. Data are mean ± SE (n = 50). Asterisks indicate significant differences in comparison with unexposed control (Student’s *t*-test, ** *p* < 0.01).

**Figure 7 plants-14-00057-f007:**
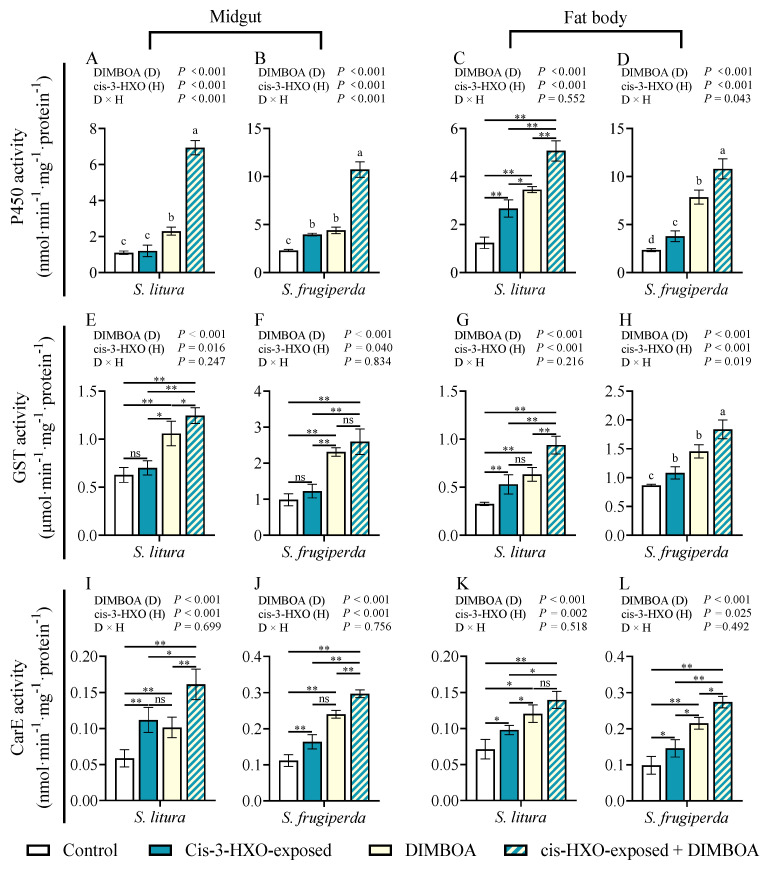
Activities of cytochrome P450 enzymes (P450s) (**A**–**D**), glutathione-s-transferase (GST) (**E**–**H**), and carboxylesterase (CarE) (**I**–**L**) in the midguts and fat bodies of 4th instars of *Spodoptera litura* and *S. frugiperda* after exposure to cis-3-hexen-1-ol (cis-3-HXO) and feeding on DIMBOA-containing diet. The larvae were exposed to cis-3-hexen-1-ol (cis-3-HXO) and 1 μg·g^−1^ DIMBOA diet for 24 h. Asterisks indicate significant differences in comparison with control (Student’s *t*-test if the interaction between HIPVs and DIMBOA was not significant, * *p* < 0.05, ** *p* < 0.01, ns, *p* > 0.05). Different letters above bars indicate significant differences among treatments (*p* < 0.05) according to two-way ANOVA with Tukey’s multiple comparison test (if the interaction between HIPVs and DIMBOA was significant).

**Figure 8 plants-14-00057-f008:**
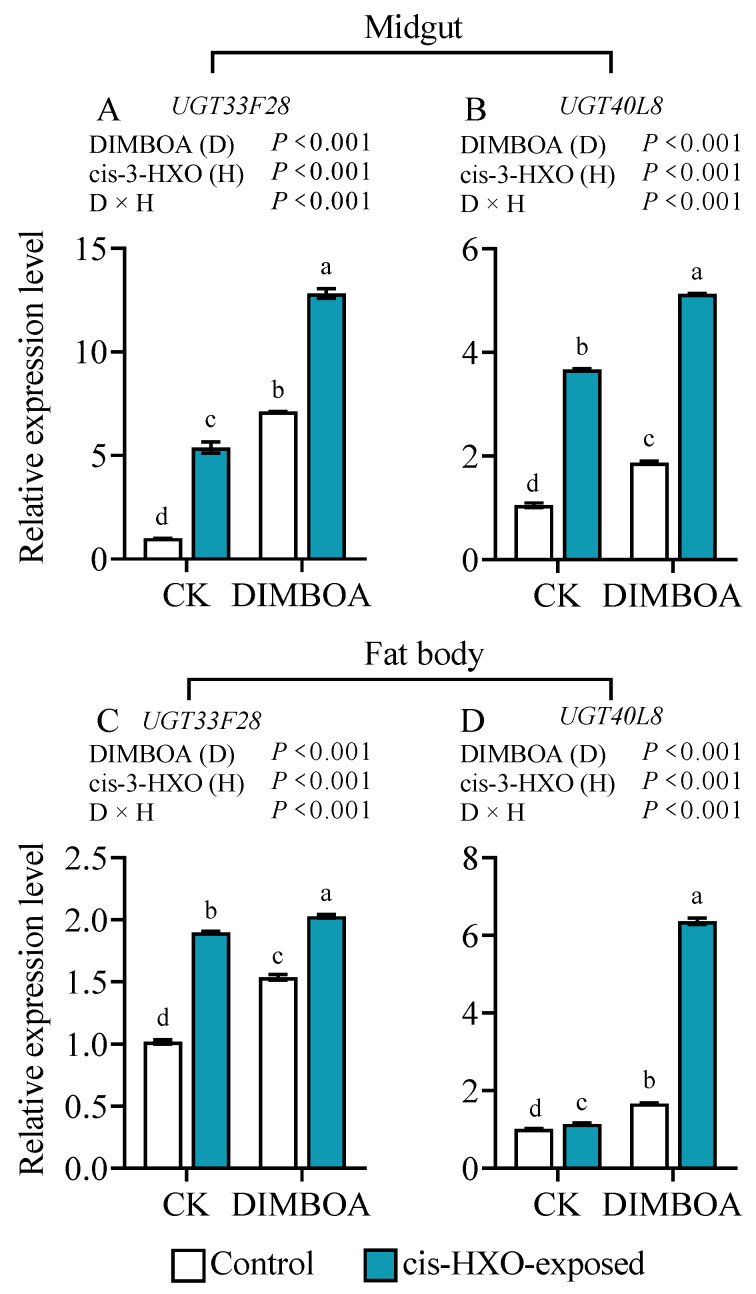
Transcript levels of *UGT33F28* (**A**,**C**) and *UGT40L8* (**B**,**D**) in the midguts (**A**,**B**) and fat bodies (**C**,**D**) of 4th instars of *Spodoptera frugiperda* after exposure to volatile cis-3-hexen-1-ol (cis-3-HXO) and feeding on DIMBOA-containing diet. The larvae were exposed to cis-3-HXO and 1 μg·g^−1^ DIMBOA diet for 24 h. The other information is described in Figure 7. Different letters above bars indicate significant differences among treatments (*p* < 0.05) according to two-way ANOVA with Tukey’s multiple comparison test.

## Data Availability

All the data analyzed during this study have been included in this article.

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
