# Peer review of "Maize Herbivore-Induced Volatiles Enhance Xenobiotic Detoxification in Larvae of Spodoptera frugiperda and S. litura"

_plants, 2024, doi:10.3390/plants14010057_

Round 1
Reviewer 1 Report
Comments and Suggestions for Authors
The manuscript by Peng Wang et al. investigates the impact of herbivore-induced plant volatiles (HIPVs) on the detoxification abilities of two polyphagous lepidopteran pests, Spodoptera frugiperda and S. litura. The results show that exposure to maize HIPVs enhances larval tolerance to various plant defenses and insecticides. This is linked to the induction of detoxification enzymes and genes, including cytochrome P450s and glutathione S-transferase. The findings suggest that HIPVs play a crucial role in improving the xenobiotic tolerance of insect larvae, supporting the idea that herbivores use HIPVs to counteract plant defenses. This manuscript presents an intriguing exploration of how herbivore-induced plant volatiles (HIPVs) affect detoxification mechanisms in Spodoptera larvae. The experimental design is promising, and the results provide important insights into pest management strategies. However, several revisions are needed to improve the clarity, statistical rigor, structure, and scientific accuracy of the manuscript.".
Major and Key Points:
- The Materials & Methods section needs to be restructured to consolidate scattered information.
- In the Results section, the authors provide general introductory statements before presenting the data. This section should focus solely on presenting the results without unnecessary background or context.
- The results section lacks essential statistical analyses (e.g., F-values, degrees of freedom, P-values), which need to be included to validate the findings.
- The manuscript contains several typos and should be carefully proofread for accuracy.
- Some of the citations in the manuscript appear to be mis-cited or mismatched. For example, reference number 16 pertains to population evolution and pesticide resistance in S. frugiperda, but it is cited in relation to the recent infestation of this species in Asia. The authors should carefully review each cited reference to ensure it directly supports the relevant claims in the manuscript.
- Some experimental details (e.g., artificial diets and compound concentrations) are not provided and should be included for reproducibility.
Miscellaneous:
The title could be shortened to 'Maize Herbivore-Induced Volatiles Enhance Detoxification in Larvae of Two Polyphagous Lepidopteran Pests' for clarity and brevity.
Alternatively, 'Maize Herbivore-Induced Volatiles Enhance Xenobiotic Detoxification in Larvae of Spodoptera frugiperda and Spodoptera litura' could be more specific and informative, highlighting the key focus of the study
L45: Is this referring to ROS (Reactive Oxygen Species)? Please clarify.
L49-50: This sentence can be rewritten as: "During co-evolution with insect herbivores, plants have developed both constitutive and inducible defenses at multiple morphological, molecular, and biochemical levels."
L52: The term "pyrethrins" needs clarification. Do pyrethrins exist in all plants? Please check reference number 7; it pertains to T. cinerariifolium.
L89: The phrase should be "in lepidopteran pests".
L63: It seems like you are referring to herbivores or insect pests. Please clarify.
L71: Reference number 16 is not about the recent infestation of S. frugiperda in Asia but about the population evolution and pesticide resistance in Spodoptera frugiperda. I suggest citing directly relevant papers rather than unrelated citations.
L97-117: This section seems more suited for the Discussion or Conclusion. I recommend the authors clarify the aim of the study here and highlight its differences from earlier studies.
L129: Provide a reference and details about the artificial diets used.
L136: What diets are referred to here? Please specify.
Section 2.2: This section needs reorganization and rewriting for clarity.
L141-148: What are the quantities of chlorogenic acid, tannic acid, and DIMBOA, as well as the insecticides chlorpyrifos and methomyl? How were the pests exposed to these compounds and pesticides? Were artificial diets used for exposure?
Section 2.3: This section should be reorganized and rewritten for clarity.
Section 2.7: More details are needed regarding which data or bioassays underwent one-way or two-way analysis.
L224-227: These sentences belong in the Materials & Methods section. They should be deleted from the Results section and only the results should be presented without M&M details.
L227-250: For the data analysis, scientific proofs (F-values, df, P-values) must be provided.
L252-253: This information is related to the Introduction or Materials & Methods sections, not here.
Figure 1: There is no mention of the use of the Student’s t-test for analyzing these variables in the Materials & Methods section. Please provide the necessary details there.
L257-259: For all the results, scientific evidence (F-values, df, P-values) must be provided.
Figure 2: All data, including those presented in the manuscript, should be shown as mean ± standard error of the mean.
L290-292: This information should be in the Introduction section, not here.
Figure 4: There is no mention of the multiple comparison test (Tukey’s test) in the Materials & Methods section. Please include the details there.
L351: Delete the phrase “in the device shown…” as it is unnecessary.
L362-364: This information belongs in the Materials & Methods section, not here.
L384-386: This sentence repeats details from the Materials & Methods section. The Results section should only present results without repeating M&M details.
L396-398: What comparison test was used for the analysis presented in Figure 8?
L405: What about microorganisms? No interactions with microorganisms are discussed. Please check reference number 21 and other relevant review papers for clarification.
L408: Please verify the cited references for accuracy.
L413: There is no mention of tobacco plants in the paper referenced as number 42. This reference pertains to tomato, not tobacco.
L416: Check the spelling of "aphid species latin name."
L419: Check for typos here.
L416: "Predators do not parasitize" — This is incorrect. Parasitoids species parasitize the hosts, not predators. Check reference number 43.
L447: Incomplete sentence. "Our results indicate that…??"
Author Response
Point by point Responses to reviewers’ comments
Responses to Reviewer 1
The experimental design is promising, and the results provide important insights into pest management strategies. However, several revisions are needed to improve the clarity, statistical rigor, structure, and scientific accuracy of the manuscript.".
Response: Thank you for your valuable comments. We have made necessary modifications and collections as suggested.
Major and Key Points:
- The Materials & Methods section needs to be restructured to consolidate scattered information.
Response: We have removed some description from Results Section to Materials & Methods section
- In the Results section, the authors provide general introductory statements before presenting the data. This section should focus solely on presenting the results without unnecessary background or context.
Response: Thank you for your suggestions. We presented 1-2 sentences of background or context to make the readers easily understand the results. We did this following examples from many papers (e.g. Wu et al. 2022. New Phytologist, 234: 1031–1046). Now we deleted some unnecessary background and simplified some.
- The results section lacks essential statistical analyses (e.g., F-values, degrees of freedom, P-values), which need to be included to validate the findings.
Response: Detailed statistical analyses are shown in the appendix.
- The manuscript contains several typos and should be carefully proofread for accuracy.
Response: The typos in the manuscript have been corrected.
- Some of the citations in the manuscript appear to be mis-cited or mismatched. For example, reference number 16 pertains to population evolution and pesticide resistance in S. frugiperda, but it is cited in relation to the recent infestation of this species in Asia. The authors should carefully review each cited reference to ensure it directly supports the relevant claims in the manuscript.
Response: The references were re-examined, and the reference 16 was replaced.
16. Sun, X. X., Hu, C. X., Jia, H. R., Wu, Q. L., Shen, X. J., Zhao, S. Y., Jiang, Y. Y., Wu, K. M. Case study on the first immigration of fall armyworm, Spodoptera frugiperda invading into China. Integ. Agri. 2021, 20, 664-672.
- Some experimental details (e.g., artificial diets and compound concentrations) are not provided and should be included for reproducibility.
Response: Details about artificial diets and compound concentrations have been added to the material method, and relevant literature was cited (Lines 112-117).
Miscellaneous:
The title could be shortened to 'Maize Herbivore-Induced Volatiles Enhance Detoxification in Larvae of Two Polyphagous Lepidopteran Pests' for clarity and brevity.
Alternatively, 'Maize Herbivore-Induced Volatiles Enhance Xenobiotic Detoxification in Larvae of Spodoptera frugiperda and Spodoptera litura' could be more specific and informative, highlighting the key focus of the study.
Response: The title has been changed as suggested by 'Maize Herbivore-Induced Volatiles Enhance Xenobiotic Detoxification in Larvae of Spodoptera frugiperda and Spodoptera litura'.
L45: Is this referring to ROS (Reactive Oxygen Species)? Please clarify.
Response: This refers to reactive oxygen species (ROS). (lines 44)
L49-50: This sentence can be rewritten as: "During co-evolution with insect herbivores, plants have developed both constitutive and inducible defenses at multiple morphological, molecular, and biochemical levels."
Response: This sentence has been rewritten as “During co-evolution with insect herbivores, plants have developed both constitutive and inducible defenses at multiple morphological, molecular, and biochemical levels”. (lines 49-50)
L52: The term "pyrethrins" needs clarification. Do pyrethrins exist in all plants? Please check reference number 7; it pertains to T. cinerariifolium.
Response: Pyrethrin is also a well-known natural defensive chemical presented in some plants. It is an active ingredient from Pyreyhrum cineriifoliun Trebr with insecticidal effect. DIMBOA, diterpenoid glycosides, and pyrethrins are only present in some plant species.
L63: It seems like you are referring to herbivores or insect pests. Please clarify.
Response: In this case, it mainly refers to herbivores.
L71: Reference number 16 is not about the recent infestation of S. frugiperda in Asia but about the population evolution and pesticide resistance in Spodoptera frugiperda. I suggest citing directly relevant papers rather than unrelated citations.
Response: Reference number 16 has been replaced by Sun, X. X., Hu, C. X., Jia, H. R., Wu, Q. L., Shen, X. J., Zhao, S. Y., ... & Wu, K. M. (2021). Case study on the first immigration of fall armyworm, Spodoptera frugiperda invading into China. Journal of Integrative Agriculture, 20(3), 664-672. (lines 511-513)
L89: The phrase should be "in lepidopteran pests".
Response:It has been corrected. (lines 89)
L97-117: This section seems more suited for the Discussion or Conclusion. I recommend the authors clarify the aim of the study here and highlight its differences from earlier studies.
Response: This section has been simplified (Lines 95-107). Some information is present in the Discussion.
L129: Provide a reference and details about the artificial diets used.
Response: Larvae were reared with an artificial diet as described by Gupta et al. (lines 133-134, Reference 27).
Reference: Gupta, G.; Birah, A.; Rani, S. Development of artificial diet for mass rearing of American bollworm, Helicoverpa armigera. Indian J. Agric. Sci. 2004, 74, 548–551.
L136: What diets are referred to here? Please specify.
Response: The diets “containing 1 μg·g-1 DIMBOA, 3 μg·g-1 chlorogenic acid and 2 μg·g-1 tannic acid”. (lines 121-122).
Section 2.2: This section needs reorganization and rewriting for clarity.
Response: This section has been rewritten (Lines 128-143).
L141-148: What are the quantities of chlorogenic acid, tannic acid, and DIMBOA, as well as the insecticides chlorpyrifos and methomyl? How were the pests exposed to these compounds and pesticides? Were artificial diets used for exposure?
Response: The details of chemicals are described in lines 128-140.
Section 2.3: This section should be reorganized and rewritten for clarity.
Response: The title of this section has been changed to “Larval Exposure to cis-3-hexene-1-ol and Its Effects on Xenobiotic Tolerance”. New information has been added (Lines 154-157).
Section 2.7: More details are needed regarding which data or bioassays underwent one-way or two-way analysis.
Response: The Shapiro-Wilk normality test and Levene's homogeneity of variance test (P>0.05) were used to test the normality of the data (P>0.05) before all statistical analyses. Under the assumption of normality and homogeneity of variance, student's t-test or analysis of variance (ANOVA) was used to compare the differences between two or more treatments (Tukey's test, P<0.05). In the case of two-factor variance, if there is an interaction between factor 1 and factor 2 (that is, P<0.05), one-way ANOVA (Tukey's test, P<0.05) is used for all treatment groups. If there is no interaction between factor 1 and factor 2 (P>0.05), then a student's t-test is performed for different levels of factor 1 or factor 2 (lines 207-215).
L224-227: These sentences belong in the Materials & Methods section. They should be deleted from the Results section and only the results should be presented without M&M details.
Response: These sentences have been moved to the Materials & Methods section (Lines 128-130).
L227-250: For the data analysis, scientific proofs (F-values, df, P-values) must be provided.
Response: To make the article readable details about statistics analyses including F-values, df, and P-values are provided in Supplemental Table S2-9.
L252-253: This information is related to the Introduction or Materials & Methods sections, not here.
Response: This information has been moved to the Materials & Methods section (Line 135).
Figure 1: There is no mention of the use of the Student’s t-test for analyzing these variables in the Materials & Methods section. Please provide the necessary details there.
Response: The use of Student’s t-test to analyze these variables was added to section 2.7 (lines 207-215).
L257-259: For all the results, scientific evidence (F-values, df, P-values) must be provided.
Response: To make the article readable details about statistics analyses including F-values, df, and P-values are provided in Supplemental Table S2-9.
Figure 2: All data, including those presented in the manuscript, should be shown as mean ± standard error of the mean.
Response: WE are sorry for the error. The data presented in Figure 2 are mean ± standard error. They have been corrected in Figure 2 legends.
L290-292: This information should be in the Introduction section, not here.
Response: This information has been deleted. It has been showed in Lines 56-61.
Figure 4: There is no mention of the multiple comparison test (Tukey’s test) in the Materials & Methods section. Please include the details there.
Response: The details about statistics analyses including F-values, df, and P-values are provided in Supplemental Table S2-9.
L351: Delete the phrase “in the device shown…” as it is unnecessary.
Response: The phrase “in the device shown…” has been deleted.
L362-364: This information belongs in the Materials & Methods section, not here.
Response: This information has been deleted.
L384-386: This sentence repeats details from the Materials & Methods section. The Results section should only present results without repeating M&M details.
Response: This sentence has been deleted.
L396-398: What comparison test was used for the analysis presented in Figure 8?
Response: Two-way ANOVA (Tukey’s test) was used for the analysis presented in Figure 8 (Line 368).
L405: What about microorganisms? No interactions with microorganisms are discussed. Please check reference number 21 and other relevant review papers for clarification.
Response: Interactions with microorganisms are not associated in this paper. The reference number 21 is OK.
L408: Please verify the cited references for accuracy.
Response: We have checked the reference and the citation is correct.
L413: There is no mention of tobacco plants in the paper referenced as number 42. This reference pertains to tomato, not tobacco.
Response: It has been corrected for tomatoes. (lines 418)
L416: Check the spelling of "aphidspecieslatin name."
Response: We checked the spelling of Acyrthosiphonpisum and it was correct.
L419: Check for typos here.
Response: It has been corrected.
L416: "Predators do not parasitize" — This is incorrect. Parasitoids species parasitize the hosts, not predators. Check reference number 43.
Response: According to reference 43, this sentence is reworded as “A recent study revealed that symbiotic bacteria present in Acyrthosiphonpisum can play a role in suppressing the emission of HIPVs in host plants to reduce the recruitment of natural enemies, leading to enhanced adaptability of herbivorous insects [43]”(lines 383-387)
L447: Incomplete sentence. "Our results indicate that…??"
Response: We deleted the incomplete sentence (lines 414).
Reviewer 2 Report
Comments and Suggestions for Authors
your work is interesting and the results are also very interesting, but some questions need to be addressed (see suggestions in the attached document)

Author Response
Point by point Responses to reviewers’ comments
Responses to Reviewer 2
106-111:is not necessary here
Response: This section has been simplified (Lines 95-107). Some information is present in the Discussion.
112-117:is not necessary here,
Read the text, can help you, https://dynamicecology.wordpress.com/2016/02/24/the-5-pivotal-paragraphs-in-a-paper/
https://dynamicecology.wordpress.com/2016/02/24/the-5-pivotal-paragraphs-in-a-paper/
Response: This section has been simplified (Lines 95-107). Some information is present in the Discussion.
118: 2. Materials and Methods
I have no doubt that all the bioassays were carried out based on robust protocols, but several of them need to be made clear in this section (so you don't have to put in the results what should be written in this section).
as there are several results presented, please separate them in a specific section here as well so as not to get confused.
Response:
122: environmental conditions, specific, please.
Response: The environmental conditions have been specified in Lines 106-108.
129: diet larvae reference too, please.
Response: Larvae were reared with an artificial diet as described by Gupta et al. (lines 133-134, Reference 27).
Reference: Gupta, G.; Birah, A.; Rani, S. Development of artificial diet for mass rearing of American bollworm, Helicoverpa armigera. Indian J. Agric. Sci. 2004, 74, 548–551.
139: how many replicates/ treatment??
Response: This experiment was triplicated (lines 127).
141: what time?? it is key for understand
Response: Weight gain of 3rd and 4th instar larvae following exposure to maize HIPVs and feeding on an artificial diet for 24 hours was measured.
144-148: consider the option of separating this part as another item (it's important to know how it was done) concentrations, exhibitions, meetings, etc.
Response: Details about artificial diets and compound concentrations have been added to the material method, and relevant literature was cited (Lines 112-117).
149-150: It's not clear how long they were exposed to the treatments, it was the entire after exposure?
Response: Weight gain of 3rd and 4th instar larvae following exposure to maize HIPVs and feeding on an artificial diet for 24 hours was measured.
162: how was the exposure???
Response: Larvae of the fourth instar of S. litura and S. frugiperda were exposed to cis-3-hexene-1-ol as shown in Figure S2. (lines 153-154).
165-166: right??, the apparatus evaluate the detoxification???
Response: As shown in Figure S2, the apparatus was developed to evaluate effect of volatile exposure of cis-3-hexene-1-ol on detoxification of S. litura and S. frugiperda larvae.
line 176 " ..pre-pupal stage were placed in containers"
line 178 "the larvae were subjected"...write sequentially how was done the bioassay.
Response: Please see 2.4 section for details (lines 168-178).
207: standardize the write, see line 192, in all document (space)
Response: We have standardized the writing. (lines 221, 223, 135).
221: there are many results displayed, which makes the work a bit confusing (see the possibility of selecting results to material supplementary.)
Response: It has been changed (Line 217)
235: figure 3???
Response: Sorry. It is figure 1. (lines 242)
236: figure 3??
Response: It is figure 1. (lines 243)
248: figure 3??
Response: It is figure 1. (lines 226)
255: these values should be showed in the material and methods,
Response: It has been moved to the material and methods.
Figure 2D: standardize the showed, please. or letter in all, or symbols * in all. please.
Response: The Shapiro-Wilk normality test and Levene's homogeneity of variance test (P>0.05) were used to test the normality of the data (P>0.05) before all statistical analyses. Under the assumption of normality and homogeneity of variance, student's t-test or analysis of variance (ANOVA) was used to compare the differences between two or more treatments (Tukey's test, P<0.05). In the case of two-factor variance, if there is an interaction between factor 1 and factor 2 (that is, P<0.05), one-way ANOVA (Tukey's test, P<0.05) is used for all treatment groups. If there is no interaction between factor 1 and factor 2 (P>0.05), then a student's t-test is performed for different levels of factor 1 or factor 2. So it has both symbols and letters in figure.
265:the figures showed letters, standardize the showed of figures
Response: The reason is described in the above Figure 2D.
290-291:introduction or discussion, here is not necessary.
Response: It has been deleted.
Figure 4G:standardize the showed, please. or letter in all, or symbols * in all. please.
Response: The reason is described in the above Figure 2D.
317-318:discussion, introduction, here is not necessary. here only results.
Response: It has been deleted.
319-323:methodology
Response: It has been deleted.
355:(A)
Response: Thank you for the suggestion. Lines 327.
356:(B)
(B)
Response: Thank you for the suggestion. Lines 327.
366:and the A, what show???
Response: The results focused on S. frugiperda. Figure 7A is on Lines 347.
Figure 7C: standardize the showed, please. or letter in all, or symbols * in all. please.
Response: The reason is described in the above Figure 2D.
Figure 8A:standardize the showed, please. or letter in all, or symbols * in all. please.
Response: The reason is described in the above Figure 2D.
Figure 8B:what is S. litura or S. frugiperda??
Response: WE only show quantitative results from S. frugiperda.
397:what is A, B,C,D???
Response: They have been specified in Figure 8 legends.
420\424\429\440/441/443:in this section not reference the figures again, discuss your results.
Response: They have been deleted.
Round 2
Reviewer 1 Report
Comments and Suggestions for Authors
I have reviewed the revised version of the manuscript by Wang et al., and the authors have responded to my previous comments and made necessary changes. However, I noticed a few issues with the manuscript that should be addressed:
- Line 385: The Latin name should be typed as Acyrthosiphon pisum, not as Acyrthosiphonpisum.
- Lines 388 and 391: It should be DIMBOA, not DIMOBA.
-
Upon reviewing the supplementary file, particularly the statistical analysis details, I found several issues that raise significant concerns regarding the validity of the results.
F-Value = 0.00 with P-Value < 0.01 (Table S2):
The F-value of 0.00 reported in the analysis cannot correspond to a P-value < 0.01. In ANOVA, the F-statistic is a ratio of the variance between groups to the variance within groups. A value of 0.00 suggests no variance between the groups, meaning the model does not explain any meaningful variation. This should result in a P-value of 1.00, not less than 0.01, because there is no evidence to reject the null hypothesis.
F-Ratios Between 0 and 1: (Table S2, S3, S5, S6, S7)
The reported F-ratios are all between 0 and 1 but are marked as statistically significant. This is highly unusual. With equal degrees of freedom for both between-group and within-group variances, an F-ratio between 0 and 1 generally implies that the model is explaining less variance than would be expected by chance. Such small F-ratios should not result in statistically significant findings, as this would suggest that there is no meaningful difference between the groups.
Potential Statistical Issues:
Because the analysis is conducted with equal degrees of freedom (likely from equal sample sizes), the observed F-ratios and their significance are particularly problematic. F-ratios near 0 or less than 1 are not typical in statistically significant findings, even with balanced designs. These discrepancies suggest that either there was an error in the statistical calculations or the data do not support the claimed conclusions.
Due to these issues, I suggest that the authors recheck their statistical analysis, especially the computation and reporting of F-values and P-values. It is important to ensure that the results align with standard statistical expectations.
Author Response
The authors have improved their manuscript based on my previous comments. However, upon reviewing the supplementary file, particularly the statistical analysis details, I found several issues that raise significant concerns regarding the validity of the results.
Response: We thank you for your valuable comments. We have made necessary modifications and collections as suggested.
F-Value = 0.00 with P-Value < 0.01 (Table S2):
The F-value of 0.00 reported in the analysis cannot correspond to a P-value < 0.01. In ANOVA, the F-statistic is a ratio of the variance between groups to the variance within groups. A value of 0.00 suggests no variance between the groups, meaning the model does not explain any meaningful variation. This should result in a P-value of 1.00, not less than 0.01, because there is no evidence to reject the null hypothesis.
Response: Two decimal places are retained for all data. This is why F-value is 0.00 reported in the analysis. Now we have added the F value.
In Table S2. Figure 1C, the F value of “S. litura Control-vs-HIPVs-exposed” was 0.00159.
In Table S2. Figure 1F, the F value of “S. frugiperda Control-vs-HIPVs-exposed” was 0.002824, and the P value was 3.11×10-12.
In Table S3. Figure 2B, the F value of “S. frugiperda Control-vs-HIPVs-exposed” was 0.003055, and the P value was 4.79×10×10-15.
In Table S7. Figure 6A, the F value of “S. litura Control-Vs-CIS-XO-exposed” was 0.00173, and the P value was 1.34103×10×10-10.
F-Ratios Between 0 and 1: (Table S2, S3, S5, S6, S7)
The reported F-ratios are all between 0 and 1 but are marked as statistically significant. This is highly unusual. With equal degrees of freedom for both between-group and within-group variances, an F-ratio between 0 and 1 generally implies that the model is explaining less variance than would be expected by chance. Such small F-ratios should not result in statistically significant findings, as this would suggest that there is no meaningful difference between the groups.
Response: We have checked the original data analysis report of all the tables, and there are some repeated errors due to replication errors (Table S5, S6 and S8), which have been corrected, but some F-values are indeed between 0 and 1 (Table S2, S3 and S7).
Potential Statistical Issues: Because the analysis is conducted with equal degrees of freedom (likely from equal sample sizes), the observed F-ratios and their significance are particularly problematic. F-ratios near 0 or less than 1 are not typical in statistically significant findings, even with balanced designs. These discrepancies suggest that either there was an error in the statistical calculations or the data do not support the claimed conclusions. Due to these issues, I suggest that the authors recheck their statistical analysis, especially the computation and reporting of F-values and P-values. It is important to ensure that the results align with standard statistical expectations.
Response: We have checked the original data analysis of all the tables, and have corrected all the errors in some places due to errors in statistical analysis.
Round 3
Reviewer 1 Report
Comments and Suggestions for Authors
I have reviewed the revised version of the manuscript by Peng Wang et al., along with the supplementary file that includes the corrected data analysis. The authors have addressed several statistical issues raised in my initial review, particularly regarding the F-values and corresponding P-values in their ANOVA results.
F-Value of 0.00 with P-Value < 0.01: In my previous review, I pointed out that an F-value of 0.00 could not correspond to a P-value < 0.01, as this would suggest no meaningful variance between groups. The authors have clarified that the F-value of 0.00 was due to rounding to two decimal places. They have now provided more precise values, including those for F-ratios between 0 and 1.
F-Ratios Between 0 and 1: I also highlighted that F-ratios between 0 and 1, when marked as statistically significant, are unusual and raise concerns about the validity of the results. The authors acknowledge that some errors occurred due to replication mistakes and have corrected the F-ratios in several tables (S2, S3, S5, S6, S7). They have now clarified the F-values for specific comparisons, such as “S. litura Control-vs-HIPVs-exposed” and “S. frugiperda Control-vs-HIPVs-exposed,” which now align with more reasonable statistical expectations.
The authors have made significant efforts to address the concerns raised, and the revisions seem to have corrected the issues identified with the statistical analysis. However, to fully assess the accuracy of the revised analysis, access to the raw data would be necessary.